# OpenReview forum: "Beyond Linear Probes: Dynamic Safety Monitoring for Language Models"
_ICLR.cc/2026/Conference — ICLR 2026 Poster_

### Official Review · Reviewer_Hn5t · 2025-10-24

**Soundness:** 3
**Presentation:** 3
**Contribution:** 3
**Rating:** 6
**Confidence:** 3

**Summary:**

This paper proposed a new dynamic safety monitoring methodology that balances the tradeoff of classification accuracy and compute cost. Instead of using a linear probe, the authors creatively use polynomial classifiers with progressive learning and early stop methodologies, achieving up to 10% improvement in accuracy over linear probes and up to 6% over MLP baselines.

**Strengths:**

1. The concept of dynamic safety monitoring with polynomial classifiers is novel. I like the idea of progressive training, early stopping, as well as symmetric CP factorization in order to keep the model weights growing linearly. All these ideas and optimization strategies provide a practical and realistic solution to safety monitoring during the deployment time
2. The experiment is comprehensive. The author did a comprehensive study on various checkpoints with different sizes, training stages, and reasoning capabilities. The author also compared the polynomial probe with 5 baseline methods, and repeated each experiment 5 times with different random seeds.
3. The writing is good. All the terminologies are clearly defined, and the plots are clear and easy to understand.

**Weaknesses:**

1. The improvement of TPC seems to be marginal. The author claimed that it can have up to 6% of the improvements compared with the MLP baselines. However, I have no idea if the observation holds for all layers in the model. Theoretically, a fair comparison should be: for each method, we identify the layer that has the best accuracy in the validation set, then test its accuracy on the test set. In this case, we should compare the performance of the TPC and MLP baseline on their corresponding "best" layer, instead of the same layer. According to Figure 6, I found that in some cases, MLP baselines have a comparable performance to the TPC. Note that, for MLP, we can still have the early stop strategy to dynamically control the compute cost. In this case, the key issue falls on the fitting function selection, and I think the author should elaborate more on this.

**Questions:**

1. In Figure 3, how do we determine the number of parameters used in TPC? IIRC the number of parameters required by TPC is dynamic and depends on the inputs?
2. How does the layer id get selected? Do we have to do a layer-wise sweep in order to get the layer with the best performance?

---

> ### Author Response · Authors · 2025-11-20
>
> We thank the reviewer for their thoughtful response. We appreciate the reviewer's assessment of the 'novel' methodology for dynamic monitoring and 'comprehensive' evaluation. We address the stated concern and both questions below:
>
> ## [W1] 'Best layer' comparison
>
> The reason we opted for line graphs at specific layers is to visualize how baselines scale at multiple partial evaluations. We agree with the reviewer that an additional comparison of test set performance at the 'layer at which the validation performance is the highest', for each baseline, is a fair setup and is also insightful.
>
> **Please find an additional complete table (Tab. 1, page 9 of the revised manuscript) where we report these requested comparisons across 4 models, 2 layers, and 2 datasets.**
>
> To summarize the results, we observe that TPCs have the highest performance on the test set for the challenging WildGuardMix dataset across all 4 models considered. For BeaverTails, its test set performance is slightly behind EE-MLPs for 3/4 base models; yet TPCs' max performance on the validation set remains the highest on 3/4 of the base models. We would respectfully suggest that these results are consistent with the claims we make in the paper (e.g., [L026], [L097], [L108]), that TPCs "compete with or outperform MLP-based baselines"; yet come with the benefit of being more of an interpretable model (through attribution of neuron combinations). Even when performing similarly, another key benefit of dynamic TPCs over regular MLPs is the ability to train a single model that scales safety at inference time, avoiding the need for $N$ separate models to be tuned and trained from scratch.
>
> We have also removed the appeal in the experimental discussion to the model-specific performance gains in the appendix, and instead present this table as a more comprehensive summary of the models (at least as evaluated statically). We are grateful to the reviewer for this helpful suggestion.
>
> ## [Q1] Calculating TPCs' parameter count
>
> The reviewer is correct that the TPC parameter count is dynamic, in the sense that one can choose the amount of compute (/parameters) to consume when processing each input. The parameter counts along the x-axis of Fig. 3 (a) are the net parameters when using a fixed order for all inputs (black) vs each input choosing its own exit via Alg. 1 (colors). Please do let us know if there are any ways we can make this more clear in the revised manuscript.
>
> ## [Q2] Layer choice
>
> We sample both a middle layer and middle-late layer from all 4 models in our experiments, following past work [1, 2]. Please also see new experiments for sweeps across **six** layers for the full `Llama-3.2-x` family in appendix **Sect. F.4** (page 23) in response to **Reviewer-1PvY**, where we see that test F1 scores are consistently above 0.80 for all layers considered, and that middle / middle-late layers produce the best scores, providing additional justification for the heuristic.
>
> We acknowledge that the need to consider multiple potential layers to probe for safety is a drawback of activation monitors, and **we add a discussion of this to the limitations section**. However, we also imagine the ability to monitor (and ensemble predictions) at multiple points in the network could allow a more reliable multi-layered 'swiss cheese-style' defense in future work.
>
> ---
>
> ## Thank you: any additional questions or concerns?
>
> We are grateful to the reviewer once more for their time and for providing a constructive review. We hope the additional experiments and discussion presented in the revised manuscript address your concerns. If any issues remain, we would be glad to offer further clarification or engage in additional discussion to strengthen the paper.
>
> Please find a summary of the new discussions and experiments in our global response at the top of OpenReview.
>
> ---
>
> ## References
>
> * [1] MacDiarmid, et al. “Simple Probes Can Catch Sleeper Agents.” Anthropic, 23 Apr. 2024, http://www.anthropic.com/news/probes-catch-sleeper-agents.
> * [2] Ju, et al. “How Large Language Models Encode Context Knowledge? A Layer-Wise Probing Study.” LREC-COLING 2024.

---

> > ### Comment · Reviewer_Hn5t · 2025-11-25
> >
> > Dear authors,
> >
> > Thanks for your response, which addresses most of my concerns. Just one more follow-up question: Given that the main advantage of TPC (compared with EE-MLP) is better interpretability. Could you explain why we need a more interpretable model for safety monitoring?

---

> > > ### Author Response · Authors · 2025-11-26
> > >
> > > **We are glad that we have addressed most of the reviewer's concerns!**
> > >
> > > Thank you for the opportunity to expand on the value of interpretability in safety monitoring. We suggest that faithfully accounting for why monitors make the decisions they do is potentially useful for both enhanced safety and user experience:
> > >
> > > * Perhaps most critically, a harmful request may end up evading the monitors. In such scenarios, we suggest it is useful when developers can understand which features caused the false negative; possibly making debugging [1] and correction easier.
> > > * When a user request is rejected, having an option to transparently explain which factors caused the refusal can be helpful and informative when interacting with an LLM [2].
> > >
> > > TPCs make safety classifications as simple linear combinations of the LLMs’ original neurons, and their interactions [3]. We show in Sect. 4.4 how this provides built-in attribution of the classifiers’ decision to specific combinations of LLM neurons. We thus believe TPCs’ model form is well-placed to offer such insights, especially when combined with sparsity constraints [4] (as we discuss in the limitations around [L531]).
> > >
> > > We would gladly include this discussion in the final manuscript if you believe this would strengthen the paper.
> > >
> > > ---
> > >
> > > **References**
> > >
> > > • [1] Wong, et al. “Leveraging sparse linear layers for debuggable deep networks.” ICML 2021.
> > >
> > > • [2] Zheng, et al. “Let Them Down Easy! Contextual Effects of LLM Guardrails on User Perceptions and Preferences.” EMNLP 2025.
> > >
> > > • [3] Dubey, et al. “Scalable interpretability via polynomials.” NeurIPS 2022.
> > >
> > > • [4] Huben, et al. “Sparse autoencoders find highly interpretable features in language models.” ICLR 2023.

---

### Official Review · Reviewer_1PvY · 2025-10-30

**Soundness:** 3
**Presentation:** 3
**Contribution:** 3
**Rating:** 6
**Confidence:** 4

**Summary:**

This article introduces the Truncuated Polynomial Classifier (TPC), which serves as a lightweight and adaptable monitoring tool. TPC builds on N-degree polynomials, making it more expressive than simple linear probes applied to activations of LLMs. The authors additionally propose a cascading defense by training TPC in a progressive manner, which allows for the usage of $n<N$ polynomial terms for the classification. After this training, the end user can keep adding polynomial terms until the prediction is confident enough (decided by a threshold $\sigma(y)\in(\tau, 1-\tau)$. Additionally, to address the issue of the exponential growth of parameters when using higher-order polynomials, the authors propose parametrizing the higher-order polynomial terms through a symmetric CP factorization. To evaluate how well the TPC performs, authors have utilized 2 datasets, WildGuard and BeaverTails, both containing safe and unsafe examples. TPC is trained on both of those datasets on top of 4 models: gemma-3-27b-it, Qwen3-30B-A3B-Base, llama-3.2-3B, and gpt-oss-20b. The authors compare their method to other probing methods, MLPs, and early exit MLPs.

**Strengths:**

The TPC is an original idea that is well motivated by the theoretical background of polynomial networks. The authors have conducted experiments on the BeaverTails and WildGuardMix datasets for training and evaluation. WildGuardMix is a highly difficult dataset even for Guard-like models, highlighting the utility of TPC. On top of that, authors have compared the performance of TPC on 4 models differing in size and kind. The authors have performed an in-depth analysis of Cascading defense, highlighting how it influences the model performance compared to training a full polynomial. These results highlight the advantage of TPC over EE-MLP. The analysis in Figure 3(b) shows the exit degree of TPC, which proves that using this cascading defense can lead to significantly smaller computations compared to the full polynomial. I find this article as an important step towards developing more robust latent-based monitors for LLMs, which could serve as an additional defense during development.

**Weaknesses:**

* Presenting most results in the form of plots, while visually appealing, makes it harder to analyse, and I think that adding a table with the main results would be beneficial for the clarity of this work.
* For each of the models, the authors have only used 2 layers. I would like to see an ablation for at least one model on how the TPC performs depending on the layer used. I don’t think that all layers have to be checked, but more would be of high benefit for this work.
* In Figure 6, we can observe that for some layers, using higher-order polynomials is worse in terms of test performance.
* I suggest that the authors provide additional information regarding the model used in figures when it’s not obvious from the legends, etc., e.g., Figures 3 and 4.
* The usage of only 1 model from each family doesn’t allow us to fully understand the scaling of TPC. It would be beneficial to analyze how TPC scales with model sizes for at least one family (e.g., use TPC on top of Llama 3.2 1B, 3B, and Llama 3.3 8B).
* The authors have also trained the TPC on the same dataset that it is evaluated and it would be beneficial to analyze the generalization of TPC to new data.
* In Figure 3(a) x-axes have a different scales.

**Questions:**

* As we can observe in Figures 6, 10, and 11, for some examples usage of higher order polynomials decreases the TPC performance. Have the authors tried to find a solution for this problem, or have any experiments regarding this problem?
* Can the authors show how high of FRR this method would achieve on benign prompts from different datasets? I'm interested in the generalization of TPC by looking at the difference in val and test set performance for some layer and model combinations.
* Can the authors provide an ablation of TPC performance dependent on the layer number?

---

> ### Author Response · Authors · 2025-11-20
> **Response [1/2]**
>
> We are grateful to the reviewer for their insightful review, suggestions, and praise for the paper's contributions. Below, we respond to each concern and question in turn, starting with the **additional experiments**:
>
> ## [W2 & W5 & Q3] Model family & layer sweep
>
> The reviewer's suggestion to compare how TPC models scale across a whole model family is an insightful one. As suggested, we take the entire Llama 3.2 family, and additionally sweep over **six** different layers in each.
>
> > **Llama-3.2-3B layer sweep** mean F1 scores (%) across 5 random seeds (28 total layers).
>
> | Split            | Method          | L8    | L10   | L12       | L16   | L20   | L24   |
> |------------------|-----------------|-------|-------|-----------|-------|-------|-------|
> | WildGuard (val)  | Linear probe    | 94.99 | 95.21 | **95.52** | 95.08 | 94.53 | 94.23 |
> | WildGuard (val)  | TPC (5th order) | 97.00 | 97.12 | **97.25** | 97.18 | 96.90 | 96.63 |
> | WildGuard (test) | Linear probe    | 84.47 | 84.49 | **84.62** | 83.18 | 82.79 | 82.67 |
> | WildGuard (test) | TPC (5th order) | 84.42 | 84.77 | **84.78** | 84.48 | 83.83 | 83.60 |
>
>
> > **Llama-3.2-1B layer sweep** mean F1 scores (%) across 5 random seeds (14 total layers).
>
> | Split            | Method          | L2     | L4     | L6        | L8        | L10    | L12    |
> |------------------|-----------------|--------|--------|----------|-----------|--------|--------|
> | WildGuard (val)  | Linear probe    | 91.34  | 92.81  | 93.76    | **93.85** | 93.49  | 93.48  |
> | WildGuard (val)  | TPC (5th order) | 95.42  | 96.25  | 96.70    | **96.87** | 96.65  | 96.50  |
> | WildGuard (test) | Linear probe    | 80.03  | 82.12  | 82.87    | **82.91** | 81.86  | 81.25  |
> | WildGuard (test) | TPC (5th order) | 80.55  | 82.91  | **84.26**| 83.42     | 83.66  | 83.02  |
>
> Please see a full discussion in **Sect. F.4** of the Appendix of the revised manuscript. Briefly, we observe that there is a slightly higher performance in the larger LLMs, yet the activation monitors perform well across the Llama-3.2 family. Secondly, we see in both cases that the best performance of activation probes lies in the middle layers (or slightly afterwards), providing some further justification for the heuristic of searching middle or later layers like past work [4,5]. We kindly note that the requested Llama-3.3-8B model is not yet open-source, and we therefore hope the full sweep over the 3.2 family will be sufficient evidence (in combination with the other 3 models of varying size already experimented with).
>
> ## [W6 & Q2] Cross-dataset generalization
>
> To assess generalization of the probes across domain, we take the main Gemma models from Figure 2 of the paper (trained on WildGuardMix's training set), and evaluate their accuracy on **three** external test sets requested by **Reviewer-9co4** [1,2,3].
>
> |                                 |           **ToxicChat** [1]           |          **HarmBench** [2]         |         **OpenAI-Moderation** [3]        |
> |---------------------------------|:-----------------------------:|:--------------------------:|:--------------------------------:|
> | Linear probe                    |             86.84             |            99.00           |               72.98              |
> | EE-MLP (5th order)              |             89.42             |            99.50           |               73.04              |
> | **TPC** (ours) (5th order) |             **91.30**             |            **100.0**           |               **74.88**              |
>
> We see that TPCs generalize well to the test sets, with strong performance throughout. We provide a full discussion in **Sect. F.2** (page 21) of the revised PDF, where we also provide explanations for the lower performance on [3] (due to distribution shifts).

---

> ### Author Response · Authors · 2025-11-20
> **Response [2/2]**
>
> ## [W1] Presentation as tables
>
> In this paper, our key interest lies in how models scale dynamically. We chose line graphs for the main results to focus attention on this aspect. However, we agree with the reviewer that an additional table of main results would provide further clarity, and are grateful for the constructive suggestion.
>
> We have updated the main paper (**Tab. 1**) to include a summary of the test set scores across all models, layers, and datasets (when considering only the full model performance without truncation or early-exits).
>
> ## [W3 & Q1] Performance drop at higher orders
>
> The reviewer is correct to note that the performance is not monotonically increasing with additional compute. We highlight that this holds not only for early-exit TPC/EE-MLP models, but also for individually trained MLPs of increasing size. We have not experimented with this thoroughly, but techniques such as distillation from adjacent/later orders' logits or initializing higher order terms' weights closer to 0 (to inherit previous order's performance exactly), could be promising steps. **We have added a discussion of this to the limitations section** (around page 10), and we are excited about future work that might explore this.
>
> ## [W4] More information about figures
>
> We thank the review for the useful suggestion; indeed the model used was not specified anywhere in Figure 3. We have made this explicitly clear in the revised caption.
>
> ## [W6] Axes scales
>
> We kindly note that the reason for the different scales on the x-axes in Figure 3 (a) is due to the fact these are the *net* parameters used across the validation and test set separately (which differ by an order of magnitude in number of samples). If the reviewer has any suggestions for how to make this more clear, we would be happy to incorporate them into the revised PDF.
>
>
> ---
>
> ## Thank you: any further questions?
>
> We appreciate the reviewer's time and the thoughtfulness of their comments. Do the additional experiments presented above resolve your concerns with the paper? If not, we would be very happy to provide further clarification or engage in more detailed discussion to improve the work.
>
> Please further find a summary of the new experiments and discussions in our global response at the top of OpenReview.
>
> ---
>
> ## References
>
> * [1] Markov et al., A holistic approach to undesired content detection in the real world
> * [2] Mazeika et al., Harmbench: A standardized evaluation framework for automated red teaming and robust refusal
> * [3] Lin et al., ToxicChat: Unveiling hidden challenges of toxicity detection in real-world user-AI conversation
> * [4] MacDiarmid, et al. “Simple Probes Can Catch Sleeper Agents.” Anthropic, 23 Apr. 2024, http://www.anthropic.com/news/probes-catch-sleeper-agents.
> * [5] Ju, et al. “How Large Language Models Encode Context Knowledge? A Layer-Wise Probing Study.” LREC-COLING 2024.

---

> > ### Comment · Reviewer_1PvY · 2025-11-24
> >
> > I thank the authors for their clarifications and additional experiments.
> >
> > First of all, I want to apologise for my typing error regarding the Llama-3.3-8B model. I intended to ask about the Llama-3.1-8B model, which is available online.
> >
> > I have an additional request regarding cross-dataset generalization, as when evaluating the model on WildGuardMix/BeaverTails, the authors show the F1 score. I suggest using the exact same metric in this cross-dataset generalization table.
> >
> > I would also like to discuss the choice of base and instruction models from other model families. It seems to me that a natural choice in such a scenario would be using the same model family in both variants, e.g., Llama-3.2-3B and Llama-3.2-3B-Instruct. Can the authors clarify their choice of base and instruct models to be from other families?

---

> ### Author Response · Authors · 2025-11-25
>
> Thanks to the reviewer for their time and constructive input! We respond to each comment:
>
> ## Additional `Llama-3.1-8B` experiments
>
> Thank you for the clarification; we have now performed full hyperparameter sweeps over six layers for the requested additional `llama-3.1-8B` model. Please find the updated results and discussion in **Sect. F.4** of the Appendix, and inline here:
>
> > **Llama-3.1-8B layer sweep** mean F1 scores (%) across 5 random seeds (32 total layers).
>
> | Split            | Method          | **L10**   | **L12**   | L16   | L20   | L24   | L30   |
> |------------------|-----------------|-------|-----------|-------|-------|-------|-------|
> | WildGuard (val)  | Linear probe    | 96.04 | **96.35** | 96.14 | 95.75 | 95.24 | 94.91 |
> | WildGuard (val)  | TPC (5th order) | 97.37 | **97.55** | 97.47 | 97.23 | 97.01 | 96.79 |
> | WildGuard (test) | Linear probe    | **85.23** | 84.84 | 84.10 | 83.17 | 83.94 | 84.06 |
> | WildGuard (test) | TPC (5th order) | **85.79** | 85.23 | 84.94 | 83.89 | 84.10 | 83.99 |
>
> Briefly, we continue to observe the relationship between the size of the base model and the maximum test set performance hold, yet the TPC probe still brings a strong $84.26$ best F1 score (%) on the test set in the smallest $1$B model. The best performance is localized to slightly earlier layers in this model from the 3.1 family, suggesting a need to sweep over layers to find best activation probe performance.
>
> Please also note our response to **Reviewer-Hn5t**, where we acknowledge that needing to consider multiple layers is a limitation of activation monitors. **We have updated the limitations to explicitly mention this ([L538])**. We are excited about future work exploring ensembles of multiple layers that may lead to even stronger, multi-layered defense.
>
>
> ## Cross-dataset metrics
>
> We apologize to the reviewer for not including the context around our choice of accuracy when presenting the cross-dataset results from Reviewer 9co4 -- not all three datasets requested by Reviewer 9co4 have examples of the 'harmless' category, skewing the the F1 score artificially, rendering it less informative as a single metric (this why we reported accuracy instead).
>
> Nonetheless, we agree with the reviewer that we ought to be consistent, and thus **we now report the F1 score, precision, recall, and FRR in the revised PDF (please see Table 3)** (with appropriate caveats regarding the F1 score). Please see **Section F.4** for a full discussion and exploration of this. Briefly: whilst we observe low precision and high recall when evaluated for two datasets labeled with different notions of what is considered 'harmful', we see **TPCs have consistently lower FRRs and precision than the baselines**. We are grateful to the reviewer for requesting additional metrics and clarifications here.
>
> ## Choice of base/instruct models
>
> With a very limited compute budget, we had a choice to make about whether we want to focus on a single model family (comparing base and instruction-tuned variants), or instead also consider different model architectures (including OpenAI's reasoning models).
>
> Given the finite resources, we ultimately concluded that demonstrating probe performance across a range of different model architectures, post-training guardrails, and reasoning tendencies has higher value than focusing solely on a single family.
>
> We are grateful for the reviewer for the chance to present the initial results across the entire `Llama-3.2` family (and `Llama-3.1-8B`), and we believe the experimental results are even more comprehensive thanks to your suggestion.

---

> > ### Author Response · Authors · 2025-11-25
> > **Thanks for the response!**
> >
> > Thanks again to the reviewer for the chance to present additional results and analysis across model families, metrics, and datasets. Might the reviewer have any other concerns standing in the way of supporting the paper's acceptance?

---

> > > ### Comment · Reviewer_1PvY · 2025-11-25
> > >
> > > I appreciate the author's response, and have no further questions as all my concerns have been addressed. Based on the additional experiments and updated article, I have decided to increase my rating to 8.

---

> > > > ### Author Response · Authors · 2025-11-25
> > > > **Thank you!**
> > > >
> > > > Thanks kindly for your support of the paper -- we're very grateful for all your additional suggestions to improve the work!

---

### Official Review · Reviewer_9co4 · 2025-10-31

**Soundness:** 3
**Presentation:** 3
**Contribution:** 3
**Rating:** 6
**Confidence:** 2

**Summary:**

This paper proposes a framework called Truncated Polynomial Classifier (TPC), which extends traditional linear probing by incorporating quadratic and higher-order interactions among neuron activations to assess the harmfulness of a given query.

The key advantage of this framework lies in its ability to scale computation dynamically at test time, similar to test-time scaling in reasoning tasks. Specifically, it allows the safety monitor to allocate more computational budget to evaluate complex queries using higher-order terms while keeping lightweight computation for easier cases.

Empirically, the method achieves consistent improvements in F1 score over previous probing-based approaches on the WildGuardMix and BeaverTails datasets.

**Strengths:**

- The paper introduces a novel idea of applying test-time scaling to activation-based safety monitoring for LLMs. To the best of my knowledge, this is the first work to apply a polynomial-based approach to safety probing.
- The experimental results are solid and clearly demonstrate the effectiveness of the proposed method compared to existing baselines.
- The presentation is clear, and the paper is well-written, making it easy to follow the motivation, methodology, and results.

**Weaknesses:**

- The paper lacks comparison to external guard models (e.g., LLM-based safety classifiers such as Llama Guard or GPT-based judges). While such models are indeed computationally heavier, safety is often a domain where additional cost is justified. Therefore, discussing or quantifying the performance gap between TPC and these more comprehensive safety systems would strengthen the paper’s practical relevance.
- Although the current experiments on WildGuardMix and BeaverTails provide reasonable validation, the safety domain demands more comprehensive evaluation. Considering the high importance of robustness in safety monitoring, it would be valuable to assess the proposed method on a wider range of safety datasets (e.g., [1], [2], [3]) to verify its generality and reliability across diverse threat scenarios.

[1] Markov et al., A holistic approach to undesired content detection in the real world

[2] Mazeika et al., Harmbench: A standardized evaluation framework for automated red teaming and robust refusal

[3] Lin et al., ToxicChat: Unveiling hidden challenges of toxicity detection in real-world user-AI conversation

**Questions:**

- What was the criterion for selecting the specific LLMs for evaluation? For example, why was the instruction-tuned version of Qwen3-30B-A3B-Base not included in the experiments?

---

> ### Author Response · Authors · 2025-11-20
>
> We thank the reviewer for their thoughtful feedback, and for praising the novelty and solid results. We provide **new experimental results in response to both concerns** below:
>
> ## [W1] Comparisons to LLM-as-monitors
>
> In response to the reviewer's request, we additionally compare to 3 external LLMs as monitors: `gpt-4o-mini`, `claude-3-haiku`, and `o3-mini`. For each sample in WildGuardMix's test set, we ask each LLM in turn to produce a binary harmfulness response. We compute the F1 scores, and compare to the results from the main paper on `gemma-3-27b-it` at Layer 40, using the full models (no truncation). The results are shown below (with rough estimates of parameter counts for the LLMs where available):
>
> | **Model**          | **F1 score** (%) | **Params** |
> |----------------|:------------------------------:|-------------------------------:|
> | Linear probe   |              86.86             |                          5,377 |
> | Bilinear probe |              87.82             |                        344,129 |
> | MLP probe      |              88.20             |                      1,382,147 |
> | EE-MLP         |              87.76             |                      1,381,918 |
> | **TPC** (ours)     |              **88.50**             |                      1,381,889 |
> | gpt-4o-mini    |              86.63             |    ≈ 8,000,000,000 |
> | claude-3-haiku |              83.24             |    ≈ 8,000,000,000 |
> | o3-mini        |              82.03             |                        unknown |
>
> We see our activation probe results (imported from the main paper) perform remarkably well against the new LLM-as-monitors in this setup, despite having orders of magnitude fewer parameters (mirroring findings by recent work [4]). Please see full details, system prompts, and discussion in new **Section F.3** of the appendix (page 22-23).
>
> ## [W2] Cross-dataset evaluation
>
> We are grateful to the reviewer for the chance to demonstrate TPC's cross-dataset performance, which we agree is an important capability to test.
>
> We take the same models as above, trained on WildGuardMix's training set, evaluating their **accuracy** at full depth (and incrementally in the revised paper) on the three suggested datasets [1,2,3].
>
>
> |                                 |           **ToxicChat**           |          **HarmBench**         |         **OpenAI-Moderation**        |
> |---------------------------------|:-----------------------------:|:--------------------------:|:--------------------------------:|
> | Linear probe                    |             86.84             |            99.00           |               72.98              |
> | EE-MLP (5th order)              |             89.42             |            99.50           |               73.04              |
> | **TPC** (ours) (5th order) |             **91.30**             |            **100.0**           |               **74.88**              |
>
> As can be seen, all models generalize well to the test sets (especially TPCs). We see a slight drop on the OpenAI-moderation dataset, which we analyze in the revised paper, attributing this to a label distribution shift in which kinds of requests are labeled as permissible. Our full results and discussion is included in **Sect. F.2** (page 21) of the revised PDF.
>
>
> ## [Q1] Criterion for selecting the LLMs
>
> As outlined on [L305], our intention is to experiment on models with a variety of existing post-training safeguards and reasoning tendencies, to explore the generality of the activation probes. It is for this reason that we chose to use the 'base model' version of Qwen3; to complement existing experiments on the instruction-tuned Gemma3 models, and reasoning model gpt-oss. We hope this answers the reviewer's question, and we would be more than happy to provide more clarifications if useful.
>
>
> ---
>
>
> ## Thank you: any additional questions or concerns?
>
> We thank the reviewer again for their time and effort in writing a helpful and constructive review. Do the experiments above address your concerns with the paper? If not, we would be happy to provide further clarifications or discussion to help improve the paper.
>
> Please also find a summary of the key changes to the paper in our global response at the top.
>
> ---
>
> ## References
>
> * [1] Markov et al., A holistic approach to undesired content detection in the real world
> * [2] Mazeika et al., Harmbench: A standardized evaluation framework for automated red teaming and robust refusal
> * [3] Lin et al., ToxicChat: Unveiling hidden challenges of toxicity detection in real-world user-AI conversation
> * [4] http://www.goodfire.ai/research/rakuten-sae-probes-for-pii-detection

---

> ### Author Response · Authors · 2025-11-25
> **Additional metrics now present in the PDF**
>
> Dear **Reviewer 9co4**,
>
> We kindly note that you will now also find additional metrics for cross-dataset performance and an updated discussion in **Section F.2** of the paper--including F1 score, precision, recall, and FRR (when defined). We initially reported accuracy alone given that not all of the requested evaluation datasets have non-zero support for the 'harmless' class, but we have since added more metrics (and appropriate caveats) following our exchange with **Reviewer 1PvY** below.
>
> Best,
>
> The authors.

---

> ### Comment · Reviewer_9co4 · 2025-11-27
>
> I appreciate the authors for the detailed response and the additional experimental results. My concern about the cross-dataset evaluation is resolved.
>
> Regarding the LLM as monitor setting, I appreciate that the authors reported results using gpt-4o-mini, claude-3-haiku, and o3-mini. However, these models are highly sensitive to the specific prompt used, which makes the evaluation less stable. If resources allow, could the authors also include results using open-sourced guard models such as meta-llama/Llama-Prompt-Guard-2-86M, hbseong/HarmAug-Guard (0.4B), meta-llama/Llama-Guard-3-8B, or google/shieldgemma-2b? These are only example options rather than a required set. There is no need to evaluate all of them. Using any model that fits within the computational budget would already make the comparison more reliable and meaningful for practitioners.

---

> > ### Author Response · Authors · 2025-11-27
> >
> > Thanks to the reviewer for your response! We are glad we have addressed your concerns about cross-dataset evaluation.
> >
> > We agree with the reviewer that additional comparisons to guard models are very valuable. We have performed the experiments for the latter two larger guard models requested -- we have updated the PDF to present the results, and include the full F1 scores on WildGuardTest inline here in response:
> >
> > | **Model**                        | **F1 score** (%) | **Params**      |
> > |----------------------------------|------------------|-----------------|
> > | Linear probe                     | 86.86            | 5,377           |
> > | Bilinear probe                   | 87.82            | 344,129         |
> > | MLP probe                        | 88.20            | 1,382,147       |
> > | EE-MLP                           | 87.76            | 1,381,918       |
> > | **TPC** (ours)                   | **88.50**        | 1,381,889       |
> > | gpt-4o-mini                      | 86.63            | ≈ 8,000,000,000 |
> > | claude-3-haiku                   | 83.24            | ≈ 8,000,000,000 |
> > | o3-mini                          | 82.03            | unknown         |
> > | Llama-Guard-3-8B                 | 76.85            | 8,030,261,248   |
> > | shieldgemma-2b (tuned threshold) | 69.90            | 2,614,341,888   |
> >
> > ---
> >
> > Thank you for the chance to show the further comparisons -- might the reviewer have any final concerns standing in the way of supporting the paper's acceptance?

---

> > > ### Comment · Reviewer_9co4 · 2025-11-28
> > >
> > > Thank the authors for conducting the additional experiments. The new results make it clear to me that the proposed method remains effective even when compared against guard models. Since the review editing option is currently unavailable, I will update my score accordingly once it becomes possible. Thank you again.

---

> > > > ### Author Response · Authors · 2025-11-28
> > > >
> > > > Dear Reviewer,
> > > >
> > > > We are pleased to hear that your final concern has been addressed, and we appreciate your increase in score. Thanks again for the constructive suggestions!

---

### Author Response · Authors · 2025-11-20
**Rebuttal summary**

We thank the reviewers for their thoughtful and constructive feedback. We are grateful that **all three reviewers highlight the novelty of the method and strength of the experimental results**.

We summarize here the **four new experimental results/comparisons** included in the revised manuscript in response to the reviewers:

- **Initial comparisons to LLM-as-monitors** (**Appendix F.3**): `gpt-4o-mini`, `claude-3-haiku`, `o3-mini`, `Llama-Guard-3-8B` and `shieldgemma-2b`. We find that TPCs (on the same layer 40 of gemma-3-27b-it) outperform or compete with these models on the WildGuard test set, whilst having significantly fewer parameters.
- **Cross-dataset evaluation** (**Appendix F.2**): We evaluate TPCs (trained on WildGuard's training set) on three requested external datasets: ToxicChat, HarmBench, and OpenAI-Moderation. Results demonstrate that the TPC model of the main paper generalizes relatively well to these test sets (and better than baseline methods).
- **Family layer sweeps** (**Appendix F.4**): We perform full sweeps across **six** layers for the full `Llama-3.2` family as requested by **Reviewer-1PvY**, confirming reasonable performance across models and extra evidence of best performance being localized to middle / middle-late layers.
- **Table of static results** (**Table 1**): As requested by **Reviewer-Hn5t** for additional clarity, we have added Table 1 to the main paper summarizing 'best layer' performance with static evaluation of models, complementing the dynamic performance captured in the existing line graphs.

Please find all additions/updates to the PDF in **green font**. In addition to making clarifications based on the reviews, we note we have updated the manuscript to include an expanded discussion of related work on cascades and ensembles. We also include new discussion throughout on recent related works combining probes with LLM-as-monitors.

We hope that the new experiments and updates to the manuscript satisfies the reviewers that there is value in this work appearing at ICLR this year.

---

### Meta-Review · Area_Chair_7Wt1 · 2026-01-13

**Summary:**

TPC aimed to extend linear probes with higher-order polynomial terms for LLM activations. This can be further evaluated progressively at test time. Thus, it can scale computation dynamically at test time which allows the safety monitor to allocate more computational budget for complex queries. The reviewers also highlighted the experiments on the BeaverTails and WildGuardMix datasets for training and evaluation. This paper received three positive reviews and two reviewers clearly indicated that their questions are addressed and mentioned about raising the scores. After reading all rebuttals, the AC thinks there were no major issues and the rebuttals had addressed almost all issues. Thus, this paper is ready to publish.

**Reviewer Concerns:**

- **Comparisons to external guard/LLM monitors and more datasets**. The authors added comparisons to multiple LLM-as-monitors and later to open-source guard models and also more cross-dataset evaluation (ToxicChat, HarmBench, OpenAI-Moderation).
- **Layer dependence** The authors added a table summarizing best validation layer performance. Authors also acknowledge layer choice is a drawback and point to new sweeps.

**Reviewer Scores:**

There are three reviewers with initial positive scores. All of them actively participated in the discussion and two of them clearly indicated their questions are addressed. After reading all rebuttals, the AC thinks there were no major issues and all initial questions are addressed. This paper is a clear accept.

---

### Decision · Program_Chairs · 2026-01-26

Accept (Poster)